# Enhancing Variant Prioritization in VarFish through On-Premise Computational Facial Analysis

**DOI:** 10.3390/genes15030370

**Published:** 2024-03-17

**Authors:** Meghna Ahuja Bhasin, Alexej Knaus, Pietro Incardona, Alexander Schmid, Manuel Holtgrewe, Miriam Elbracht, Peter M. Krawitz, Tzung-Chien Hsieh

**Affiliations:** 1Institute for Genomic Statistics and Bioinformatics, University Hospital Bonn, Rheinische Friedrich-Wilhelms-Universität Bonn, 53127 Bonn, Germany; meghna@uni-bonn.de (M.A.B.); knausa@uni-bonn.de (A.K.); incardon@uni-bonn.de (P.I.); schmida@uni-bonn.de (A.S.); pkrawitz@uni-bonn.de (P.M.K.); 2Core Unit for Bioinformatics Data Analysis, Medical Faculty, University of Bonn, 53127 Bonn, Germany; 3CUBI—Core Unit Bioinformatics, Berlin Institute of Health, 10117 Berlin, Germany; manuel.holtgrewe@bih-charite.de; 4Institute for Human Genetics and Genomic Medicine, Medical Faculty, RWTH Aachen University, 52062 Aachen, Germany; mielbracht@ukaachen.de

**Keywords:** variant prioritization, facial imaging analysis, next-generation phenotyping, rare diseases, exome sequencing analysis

## Abstract

Genomic variant prioritization is crucial for identifying disease-associated genetic variations. Integrating facial and clinical feature analyses into this process enhances performance. This study demonstrates the integration of facial analysis (GestaltMatcher) and Human Phenotype Ontology analysis (CADA) within VarFish, an open-source variant analysis framework. Challenges related to non-open-source components were addressed by providing an open-source version of GestaltMatcher, facilitating on-premise facial analysis to address data privacy concerns. Performance evaluation on 163 patients recruited from a German multi-center study of rare diseases showed PEDIA’s superior accuracy in variant prioritization compared to individual scores. This study highlights the importance of further benchmarking and future integration of advanced facial analysis approaches aligned with ACMG guidelines to enhance variant classification.

## 1. Introduction

Approximately 6% of the worldwide population is affected by rare diseases [1]. Whole exome sequencing (WES) has been proven to facilitate the diagnosis of rare diseases [2]. However, analyzing the tremendous variants generated by WES has become an issue. Therefore, efficiently prioritizing the variants relies on algorithms, databases, and annotations to assess and rank variants based on many parameters, including the predicted impact on protein structure and function, population frequency, and associations with established diseases.

In addition to analyzing the properties of the variants, utilizing clinical phenotypes to help the diagnosis is crucial. As patients’ clinical phenotypes can be documented by the Human Phenotype Ontology (HPO) terminology [3], many computational approaches have been developed based on HPO terms to diagnose rare diseases [4,5,6,7,8,9,10,11,12,13,14,15,16]. In addition, many rare diseases often present a characteristic pattern of facial features called “facial gestalt”. With the recent advances in computer vision, the next-generation phenotyping (NGP) approaches that analyze a patient’s frontal image have proven capable of diagnosing patients with rare disorders [17,18,19,20,21,22,23,24,25,26]. The Prioritization of Exome Data by Image Analysis (PEDIA) study has demonstrated that integrating facial and clinical feature analysis into variant prioritization significantly improves performance [27]. However, the facial analysis approach employed in the original PEDIA study, namely DeepGestalt [21], was facilitated by Face2Gene, a proprietary tool that poses challenges for seamless integration. In 2023, GestaltMatcher [25], the extension to DeepGestalt, released the open-source version [28] that trains on GestaltMatcher Database [29] compliant with the findability, accessibility, interoperability, and reusability (FAIR) principles. This update offered an on-premise solution for conducting facial analysis. Hence, the critical aspect lies in effectively facilitating the integration of these tools into the variant prioritization process.

This study showcased how we integrated facial analysis (GestaltMatcher) and feature analysis (CADA [12]) into VarFish [30], an open-source framework designed for variant analysis (Figure 1). VarFish provides a user-friendly interface and visualization tools that facilitate efficient exploration and interpretation of variants, enabling analysts to navigate complex genomic data easily. We further performed performance benchmarking on 163 patients enrolled in TRANSLATE-NAMSE (TNAMSE), a German national rare disorder project [26]. Each patient contained HPO terms, a facial image, and exome data. The integration of VarFish with GestaltMatcher exemplifies the capability to analyze any medical images within a variant analysis platform. It is essential for users seeking the on-premise solution because of privacy concerns.

## 2. Materials and Methods

### 2.1. Overview

This research focused on a cohort of 163 patients with rare diseases from the TNAMSE project who consented to have their facial images evaluated using GestaltMatcher. VarFish provided a CADD score (molecular level) [32,33], CADA score (feature level) [24], GestaltMatcher score (facial level) [25], and PEDIA score [27]. PEDIA combined scores from CADD, CADA, and GestaltMatcher scores using a support vector machine. The step-by-step integration process comprised the following stages: initiating the services individually, configuring VarFish settings, enabling face-based prioritization, and incorporating PEDIA-based prioritization within VarFish. These processes facilitated data transmission among services, aiding in variant filtering and displaying associated scores in a resulting variants table. The evaluation process entailed analyzing data from 163 patients within the cohort by exporting tables from VarFish and identifying the position of the disease-causing gene within sorted PEDIA scores. The performance evaluation included plotting the percentage of cases in which the disease-causing gene appeared within the top 1 to top 100 genes.

### 2.2. Cohort Description

A total of 163 patients diagnosed with rare diseases who provided written informed consent for their facial images to be evaluated using GestaltMatcher and for the results to be utilized in exome variant interpretation were selected from the TNAMSE project. We collected the clinical description encoded in HPO terms, a frontal facial image, and exome sequencing data for each patient. In total, 64 different monogenic disorders were identified. Benchmarking was conducted on the main and validation cohorts from the TNAMSE project. The main cohort comprised 94 patients, with 194 being pediatric cases and 30 adults, representing cases from the actual prospective study of TNAMSE. After the three-year recruitment period, an additional 69 patients were enrolled to form the validation cohort. For simplicity in benchmarking, we combined the main and validation cohorts into a single cohort for analysis in this paper.

### 2.3. Prioritization Approaches

VarFish [30] serves as the analysis platform where sequencing data are uploaded in Variant Call Format (VCF). CADD scores [32] are utilized to predict the deleteriousness of variants, with the input for the CADD service consisting of chromosome number, position, reference, and alternate allele (e.g., 1-25893242-TG-CA). Moreover, VarFish offers a user-friendly graphical interface (GUI) for inputting HPO terms. Furthermore, CADA scores [27] predict pathogenicity by leveraging phenotypic information, with HPO terms serving as inputs for the CADA service. Gestalt scores predict syndromic similarity, with the input for the GestaltMatcher service being the frontal photo of the patient. Ultimately, PEDIA combines these scores to produce a single combined score per gene, utilizing a support vector machine trained on CADD, CADA, and Gestalt scores.

Integrating multiple prioritization tools presents challenges stemming from variations in algorithms, data formats, and the necessity to harmonize diverse outputs for compatibility across tools. To tackle this issue, we seamlessly integrated GestaltMatcher into VarFish using iFrames. iFrames offers a versatile and streamlined approach for embedding external content into web pages, thereby improving functionality, user experience, and facilitating modularity and reusability.

### 2.4. Step-by-Step Setup

The integration between VarFish and other independent tools such as CADA, GestaltMatcher, and PEDIA was executed through the following steps:Initialization of Services: VarFish, CADD, CADA, GestaltMatcher, and PEDIA are initiated as separate web services. For instance, these services can be initialized on the same machine but on different ports. Instructions for starting each service can be found in the Appendix A.Configuration in VarFish: VarFish’s settings file is configured to include the URLs for the CADD, CADA, GestaltMatcher, and PEDIA web services. This ensures seamless communication and interaction between VarFish and the aforementioned tools. These tools can be hosted either in the same machine for the on-premise solution or accessible via the web services provided by the inventors.Integration of GestaltMatcher into Prioritization:
3.1.The user activates GestaltMatcher within VarFish. The face sender module from the PEDIA middleware is embedded as an iFrame in the prioritization page of VarFish.3.2.Upon selecting the frontal image of the patient for the case and submitting it, the image is transmitted via the POST method exposed by the REST API endpoint of the GestaltMatcher web service.3.3.After receiving a successful response from GestaltMatcher, the suggested gene list along with scores is relayed to the parent window of VarFish.3.4.Additionally, the file name of the last photo successfully submitted to GestaltMatcher is transmitted back to VarFish. This communication from the embedded child frame to the parent window is facilitated using the window.postMessage method.3.5.A listener is incorporated into the prioritization page of VarFish to capture message events sent from the iFrame. The received data are then stored in the variant query store of VarFish. This process ensures that the patient image does not require re-uploading when the case is reopened in VarFish, as it only needs to be submitted once per case.3.6.Subsequently, when the user performs filtering, the resulting variants table displays the Gestalt scores obtained from the last image submitted to GestaltMatcher.Enabling PEDIA-based prioritization: within the prioritization page of VarFish, it automatically triggers phenotype-based prioritization using the CADA algorithm. Furthermore, it activates variant pathogenicity-based prioritization utilizing CADD scores, which predict the deleteriousness of the variants.

### 2.5. Automation

To streamline the analysis steps with a huge number of patients and be able to reproduce the results efficiently, we employed the Automa tool (https://www.automa.site/, accessed on 20 November 2023), seamlessly integrated as a browser extension. Automa, a specialized automation tool crafted for the efficient execution of repetitive tasks in web browsers, empowers users to construct scripts or workflows. These scripts facilitate programmatic interactions with web browsers, automating tasks seamlessly. In our study, Automa played a pivotal role in automating a series of tasks, including opening a case, configuring diverse filtering criteria, uploading patient photos, enabling prioritization, applying specific filters, and exporting the resulting table comprising variants and scores. The detailed workflow for these tasks using Automa is accessible at (https://github.com/igsb/pedia-middleware/blob/master/workflow.automa.json, accessed on 4 February 2024).

## 3. Results

### 3.1. Step-by-Step Analysis

The user uploads a case or opens an existing case within VarFish. Within the filtering variants page, the user can customize filter settings according to preferences, such as a population frequency threshold of 1% (Figure 2), specific quality criteria (Figure 3), and variant impact (Figure 4).

2.CADA scores are acquired from the CADA web service by transmitting clinical features in HPO terms.3.Subsequently, CADD scores are obtained from the CADD web service by forwarding the filtered variants. The highest CADD score is chosen for each gene.4.Upload the patient’s facial image to obtain Gestalt scores calculated by GestaltMatcher (Figure 5).5.After checking the “Enable PEDIA-based prioritization” button and clicking “Filter & Display,” these scores (CADA, CADD, and Gestalt) are then dispatched to the PEDIA web service via the REST API endpoint to procure PEDIA scores per gene.6.In the resulting variants table (Figure 6), PEDIA scores are displayed in a distinct column alongside CADA, CADD, and Gestalt scores. Variants associated with genes having higher PEDIA scores are prioritized accordingly.

### 3.2. Visualizing Results in VarFish

The user can sort the variants based on the different scores (Figure 6), allowing for prioritization of the candidate list. Additionally, they can export the resulting table as an Excel or TSV (Tab-Separated Values) file for further analysis and documentation. This functionality provides flexibility and convenience in managing and presenting the variant data. For example, the patient presenting in Figure 6 and Figure 7 was molecularly diagnosed with Impaired Intellectual Development and Distinctive Facial Features with or without Cardiac Defects (OMIM: 616789). When we sorted the variants by PEDIA scores, the disease-causing mutation in *MED13L* was correctly ranked in the top position (Figure 6). Moreover, visualizing the distribution of PEDIA scores by Manhattan plot (Figure 7), we can see that the PEDIA score of *MED13L* is higher than zero and clearly on top of the other genes.

### 3.3. Performance Comparison

We analyzed the 163 cases in the TNAMSE study. We plotted the percentage of cases concerning the top-ranking genes to visualize the performance (Figure 8). Our study encompassed the assessment of top-1, top-10, and top-100 accuracy within both primary and validation cohorts. We compared the performance of using CADD, CADA, GestaltMatcher solely, CADA plus CADD, and PEDIA. Figure 8 shows that with the PEDIA score, the disease-causing gene of 57.67% of patients was ranked in the top one position, and 83.44% were ranked in the top ten. In top ten accuracy, PEDIA is 7.37% higher than using CADD plus CADA (molecular and feature scores), and the PEDIA approach outperformed all the other score settings.

## 4. Discussion

This study showcases the integration of GestaltMatcher and PEDIA within VarFish, an open-source variant analysis framework. However, the previous version of the PEDIA approach relied on the support of DeepGestalt from the Face2Gene platform run by FDNA Inc. for the gestalt scores. The non-open-source nature of DeepGestalt posed challenges for its integration into any variant prioritization platform. To address this, we replaced DeepGestalt with an open-source version of GestaltMatcher [28] provided by the Institute for Genomic Statistics and Bioinformatics at University Hospital of Bonn, making the models accessible to the research community. Facial images are considered sensitive data, and many patients are reluctant to consent to data transfer outside of the hospital. Therefore, an on-premise solution is essential for such integrations. This initiative enabled GestaltMatcher to function as an on-premise solution, allowing users to analyze patient images without requiring data transfer consent. The successful integration of GestaltMatcher and PEDIA within VarFish serves as an exemplary model for developers seeking on-premise solutions. It demonstrates how facial analysis can seamlessly be integrated into various platforms, facilitating broader adoption and utilization across domains.

In Figure 6 and Figure 7, the disease-causing mutation was ranked at the top one position, despite it having relatively low CADD and CADA scores. Upon examining the entire cohort comprising 163 patients, it was found that the disease-causing gene of 57.67% of patients was ranked at the top one position, while that of 83.44% was ranked within the top ten. These findings underscore the significant performance of PEDIA compared to other scores, indicating its efficacy in facilitating variant prioritization.

This study does not assert that the current PEDIA approach, which combines CADD, CADA, and Gestalt scores, represents the state-of-the-art method for prioritizing exome variants. Rather, the PEDIA study merely demonstrates that integrating facial image analysis with feature and molecular scores can significantly enhance performance. In the future, further benchmarking of various combinations of available tools, such as LIRICAL [11], AMELIE [6], and Exomizer [32,33], could provide additional insights. Moreover, considering that the GestaltMatcher Database adheres to the FAIR (Findable, Accessible, Interoperable, and Reusable) principles, we anticipate the emergence of numerous advanced facial analysis approaches in the near future.

Incorporating the PEDIA not only enhances performance, but also minimally impacts runtime and hardware requirements. GestaltMatcher operates efficiently without the need for GPU installation, analyzing an image in approximately seven seconds. With GPU support, this analysis time can be reduced to around three seconds. Additionally, the CADA analysis also completes within a few seconds. However, assessing the overall runtime of VarFish is challenging due to variations in annotation databases and filtering parameters. Typically, filtering exome data for a single patient takes only a few minutes. Notably, VarFish recommends a minimum of 16 CPU cores, 64 GB RAM, and 300 GB storage. As such, any increase in the overall runtime of the PEDIA process is likely negligible. It also indicates that PEDIA can be implemented in any platform.

In the future, it will be crucial for facial analysis to provide additional evidence supporting the ACMG (American College of Medical Genetics and Genomics) variant classification guidelines [34,35]. For instance, integrating features that align with the PP4 criteria, which assesses phenotype match (i.e., whether a patient’s phenotype or family history is highly specific for a disease with a single genetic etiology), could be valuable. This kind of evidence would be particularly beneficial in addressing the challenge of Variants of Unknown Significance (VUS) by potentially reclassifying them as likely pathogenic, thus providing more precise clinical guidance.

## 5. Conclusions

In conclusion, this study effectively integrates GestaltMatcher and PEDIA within VarFish, overcoming challenges related to data privacy and closed-source components. PEDIA demonstrates strong performance in prioritizing disease-causing genes, highlighting its potential in variant prioritization. Future advancements in facial analysis supporting ACMG guidelines are essential for improving clinical decision making, particularly regarding VUS.

## Figures and Tables

**Figure 1 genes-15-00370-f001:**
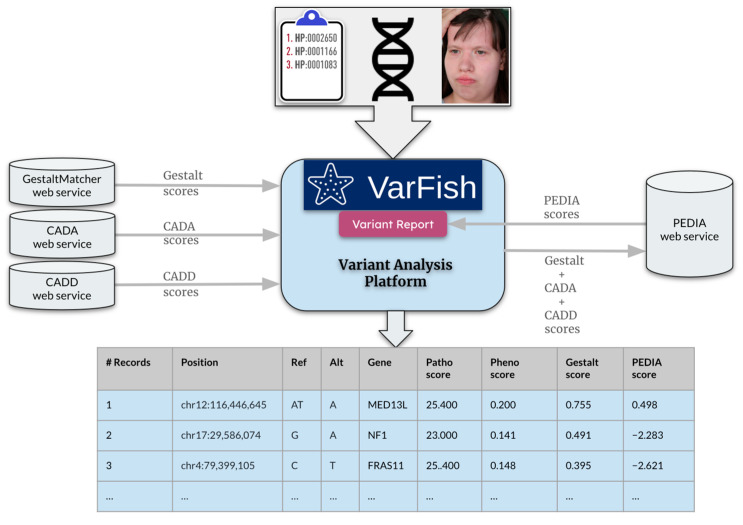
Integrating VarFish and PEDIA: Illustration depicting the seamless integration process between VarFish and PEDIA for variant prioritization. Sequencing data in VCF format is imported into VarFish, where filtering is applied. Patient images, not supported by VarFish, are uploaded via a separate web service embedded within VarFish. Gestalt scores per gene are derived using the GestaltMatcher web service. VarFish automatically retrieves CADA scores per gene from the CADA web service and CADD scores for variants from the CADD service. These scores are combined and sent to the PEDIA web service to compute the combined PEDIA score, which is displayed in VarFish’s user interface for variant prioritization. Filtered variants and their scores can be exported as a comprehensive report from VarFish. In this patient, we sorted the variants by PEDIA score in descending order, and the disease-causing mutation in *MED13L* was correctly ranked at the first position. Use of the patient’s image is consented to for publication [31].

**Figure 2 genes-15-00370-f002:**
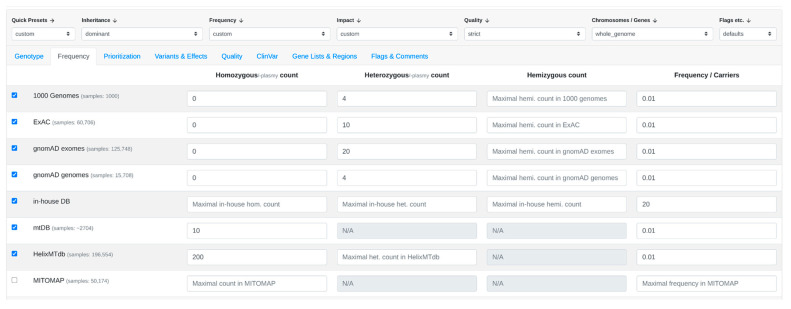
Frequency filter settings in VarFish. Every row is a different dataset. The user can set the allele frequency filter on the public and in-house datasets.

**Figure 3 genes-15-00370-f003:**
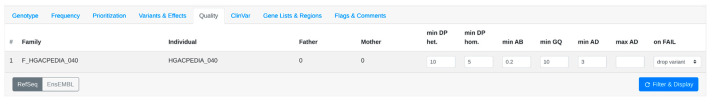
Quality filter settings in VarFish.

**Figure 4 genes-15-00370-f004:**
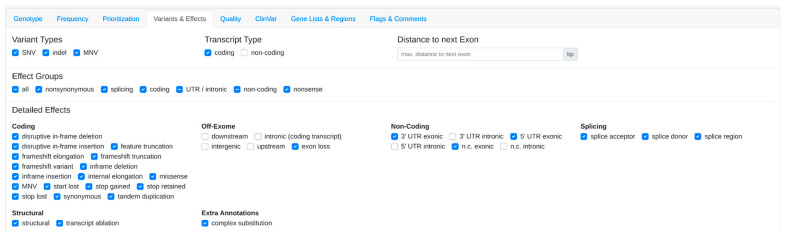
Variants and Effects filter settings in VarFish.

**Figure 5 genes-15-00370-f005:**
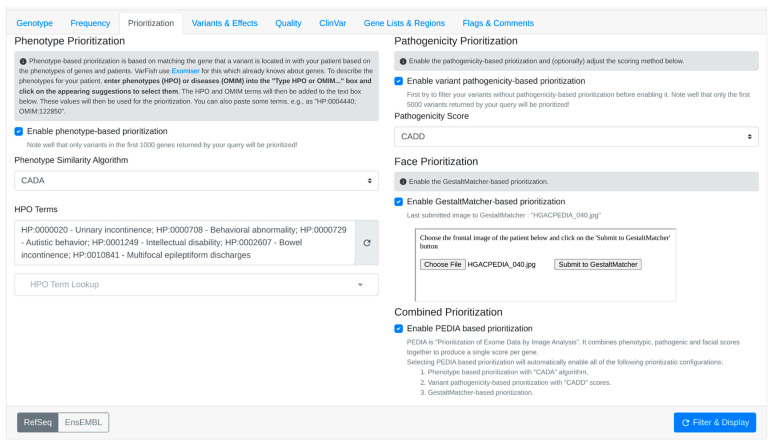
Enabling the PEDIA-based prioritization requires the following settings: phenotype prioritization using the CADA algorithm, variant pathogenicity-based prioritization using CADD, and face-based prioritization with images successfully submitted to GestaltMatcher.

**Figure 6 genes-15-00370-f006:**
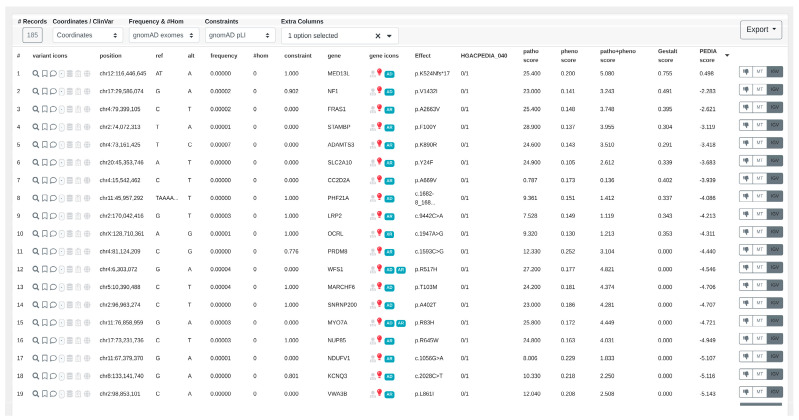
The results table of VarFish shows the variants sorted/ranked by the PEDIA scores after the filtering is performed. The CADA, CADD, and Gestalt scores are also shown in the table to increase the explainability of how the final PEDIA score was obtained. The disease-causing mutation in *MED13L* was correctly ranked at the first position.

**Figure 7 genes-15-00370-f007:**
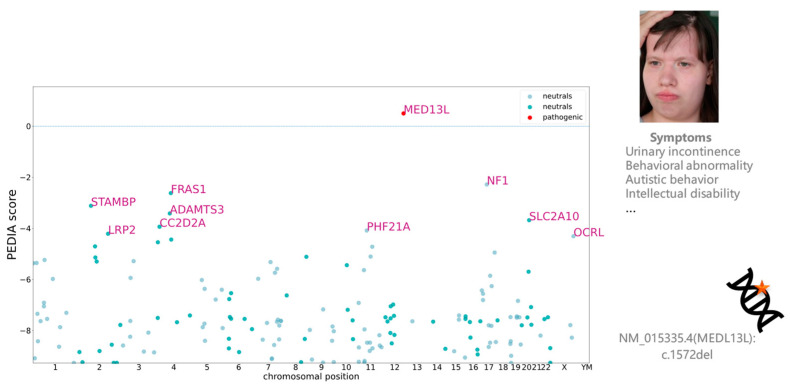
This is an illustrative case where the disease-causing gene achieves the highest PEDIA score, confirming the diagnosis of Impaired Intellectual Development and Distinctive Facial Features with or without Cardiac Defects (OMIM: 616789). *MED13L* was ranked first in this case, providing molecular confirmation of the diagnosis.

**Figure 8 genes-15-00370-f008:**
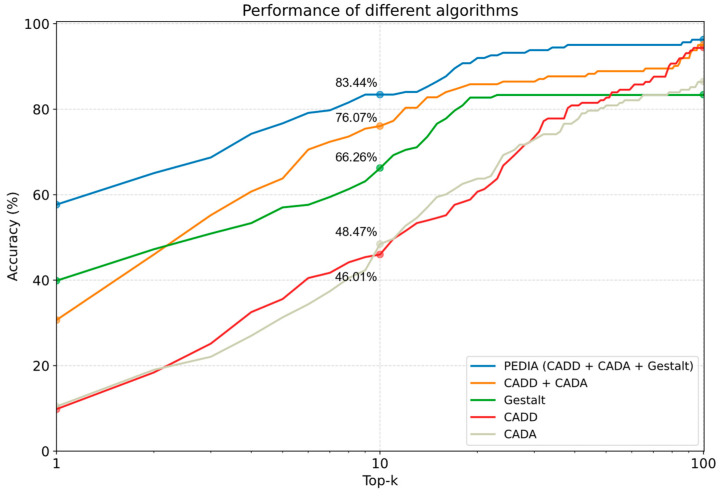
Specifically, on the y-axis, we represented the percentage of cases wherein the disease-causing gene appeared at positions ranging from top 1 to top 100. The x-axis denoted the top k genes. The Gestalt score utilizes facial image analysis through GestaltMatcher. The CADD score is derived from molecular pathogenicity assessments. CADA relies on clinical feature analysis. PEDIA integrates these three scores for variant prioritization.

## Data Availability

The source code for this project is accessible on GitHub at the following URLs: https://github.com/ahujameg/varfish-server, accessed on 12 March 2024 and https://github.com/igsb/pedia-middleware, accessed on 1 March 2024 (v1.0). The documentation can be found in PEDIA-middleware ReadTheDocs (https://pedia-middleware.readthedocs.io/en/latest/index.html, accessed on 29 February 2024). The data supporting this research are available upon request. Please contact Dr. Tzung-Chien Hsieh to request access to the data.

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
