# Peer review of "Enhancing Variant Prioritization in VarFish through On-Premise Computational Facial Analysis"

_genes, 2024, doi:10.3390/genes15030370_

Round 1
Reviewer 1 Report
Comments and Suggestions for Authors
The paper "Enhancing Variant Prioritization in VarFish through On-Premise Computational Facial Analysis" by Bhasin et al. is well-written. It is particularly commendable for demonstrating, with practical examples, the benefits of developing software that adheres to the FAIR principles in the domain of rare disease diagnosis, namely the provision of open-source software. Furthermore, the impact of on-premise computational facial analysis through the open-source version of GestaltMatcher is well summarized. For these reasons, I believe this paper will strongly capture the interest of the readers of the Genes journal.
However, it is desirable to address the following minor comments before publication:
- 1) Lines 234-237 are somewhat unclear. Does GestaltMatcher here refer to a non-open-source version integrated into Face2Gene? If so, please revise the text accordingly.
2) Since the abbreviation "VUS" is introduced at line 268, please use the term VUS at line 276.
Author Response
Dear Reviewer,
We would also like to thank the thorough evaluation of our work and the constructive comments. We have adapted all the comments and showed them point-by-point below.
Point-by-point reply:
- Lines 234-237 are somewhat unclear. Does GestaltMatcher here refer to a non-open-source version integrated into Face2Gene? If so, please revise the text accordingly.
Answer: Thank you for pointing it out. GestaltMatcher was developed by Institute for Genomic Statistics and Bioinformatics (IGSB) at University Hospital of Bonn and FDNA Inc. in 2022. Both parties provide GestaltMatcher as a service. The difference is that FDNA trained the method on their internal database. The training data and models are not shareable with the research community. Although understandable, the non-shareable data and models are not good to the scientific community and pose the challenge of connecting the facial analysis approach to any downstream analysis platform. Therefore, we (IGSB) provide the open-source version published in Current Protocol (T.-C. Hsieh, Lesmann, and Krawitz 2023), and provide the models and training data from GestaltMatcher Database (GMDB). The applicants need to submit the consent form and a research proposal to the GMDB committee to review and decide whether we grant access to the training data and models to protect the usage of this sensitive data.
Therefore, the GestaltMatcher stated in this article is provided by the open-souced version from IGSB, not the version from the Face2Gene platform. We changed lines 234-237 to the following version to better explain it:
“However, the previous version of PEDIA approach relied on the support of DeepGestalt from Face2Gene platform run by FDNA Inc. for the gestalt scores. The non-open-source nature of DeepGestalt posed challenges for its integration into any variant prioritization platform. To address this, we replaced DeepGestalt with an open-source version of GestaltMatcher (T.-C. Hsieh, Lesmann, and Krawitz 2023) provided by the Institute for Genomic Statistics and Bioinformatics at University Hospital of Bonn, making the models accessible to the research community.”
- Since the abbreviation "VUS" is introduced at line 268, please use the term VUS at line 276.
Thank you for your suggestion. We fixed it accordingly.
Thank you very much.
Best regards,
Dr. Tzung-Chien Hsieh
Reviewer 2 Report
Comments and Suggestions for Authors
The authors present in their manuscript "Enhancing Variant Prioritization in VarFish through On-Premise Computational Facial Analysis" a compelling integration of facial analysis and Human Phenotype Ontology analysis into the VarFish framework. While the manuscript is well-crafted, I have some suggestions to enhance its impact and applicability.
The manuscript would benefit from an expanded discussion on the system's performance and scalability, especially when applied to extensive genomic datasets. Also, if possible, use a Dockerfile or something else to increase the portability of the application.
Given the system's complexity, a detailed guide or documentation (e.g., via ReadTheDocs) would significantly aid new users in navigating its functionalities. This addition could enhance user accessibility and encourage wider use of your system.
Related to the GitHub repository:
It is advisable not to share the Django SECRET_KEY within the repository. Utilizing Python's native "random" function to generate unique keys for each instance can bolster the system's security; even if this is to run locally, having all instances with the same SECRET_KEY is not recommended.
Including a requirements.txt file would accommodate diverse user preferences for environment management tools, such as Conda or Pipenv. This approach also ensures that users operate with tested and functional versions of dependencies; this can be attained with "pipenv lock -r > requirements.txt".
Implementing version tagging for commits in the repository can streamline the tracking of updates and functional versions, facilitating better version control and reproducibility. This means the functional version to be released should accommodate the first tag release.
Author Response
Dear Reviewer,
We would also like to thank the thorough evaluation of our work and the constructive comments. We have adapted all the comments and showed them point-by-point below.
- The manuscript would benefit from an expanded discussion on the system's performance and scalability, especially when applied to extensive genomic datasets. Also, if possible, use a Dockerfile or something else to increase the portability of the application.
Answer: Thank you for bringing this to our attention. We've made enhancements to our repository by including the Dockerfile and docker-compose files, streamlining the setup and execution of our service.
Furthermore, we've incorporated a new paragraph in the Discussion section to delve deeper into the system requirements and runtime considerations. Given the focus of our article on integrating facial analysis and PEDIA into an existing variant prioritization platform, we've aimed to address what is essential and what additional factors may be influenced.
Notably, both GestaltMatcher and PEDIA analysis operate efficiently without requiring a GPU, completing image analysis in approximately seven seconds. Considering the overall runtime for analyzing an exome in VarFish is only a few minutes, the runtime of PEDIA can be deemed negligible in comparison. However, it's important to acknowledge the challenge in accurately assessing the overall runtime of VarFish due to variations in hardware settings and annotation databases used in the filtering process.
“Incorporating the PEDIA not only enhances performance, but also minimally impacts runtime and hardware requirements. GestaltMatcher operates efficiently without the need for GPU installation, analyzing an image in approximately seven seconds. With GPU support, this analysis time can be reduced to around three seconds. Additionally, the CADA analysis also completes within a few seconds. However, assessing the overall runtime of VarFish is challenging due to variations in annotation databases and filtering parameters. Typically, filtering exome data for a single patient takes only a few minutes. Notably, VarFish recommends a minimum of 16 CPU cores, 64 GB RAM, and 300 GB storage. As such, any increase in the overall runtime of the PEDIA process is likely negligible. It also indicates that PEDIA can be implemented in any platform.”
- Given the system's complexity, a detailed guide or documentation (e.g., via ReadTheDocs) would significantly aid new users in navigating its functionalities. This addition could enhance user accessibility and encourage wider use of your system.
Answer: Thank you very much for the suggestion. We add the ReadTheDocs (https://pedia-middleware.readthedocs.io/en/latest/index.html) to improve the documentation. We also add the following sentence in the Data availability statement in the manuscript.
“The documentation can be found in PEDIA-middleware ReadTheDocs (https://pedia-middleware.readthedocs.io/en/latest/index.html).”
- It is advisable not to share the Django SECRET_KEY within the repository. Utilizing Python's native "random" function to generate unique keys for each instance can bolster the system's security; even if this is to run locally, having all instances with the same SECRET_KEY is not recommended.
Answer: Thank you very much for the suggestion. We replace the key with a default string required to be changed by the user. We added the following instruction in Readme (https://github.com/igsb/pedia-middleware?tab=readme-ov-file#update-secret-key).
Update secret-key
Generate a Django secret key using following command:
python manage.py shell -c "from django.core.management.utils import get_random_secret_key; print(get_random_secret_key())"
Paste the newly generated key in the DJANGO_SECRET_KEY enviornment variable in the pedia_mid\settings.py file which contains the following line:
SECRET_KEY = env("DJANGO_SECRET_KEY", default="ChangeMe!!")
- Including a requirements.txt file would accommodate diverse user preferences for environment management tools, such as Conda or Pipenv. This approach also ensures that users operate with tested and functional versions of dependencies; this can be attained with "pipenv lock -r > requirements.txt".
Answer: Thank you very much for the suggestion. We added requirement.txt file and Pipenv in the repository to improve the management of the environment.
- Implementing version tagging for commits in the repository can streamline the tracking of updates and functional versions, facilitating better version control and reproducibility. This means the functional version to be released should accommodate the first tag release.
Answer: Thank you very much for the suggestion. We add the tag v1.0 in the repository and also state it in the Data availability statement in the manuscript.
“The source code for this project is accessible on GitHub at the following URLs: https://github.com/ahujameg/varfish-server and https://github.com/igsb/pedia-middleware (v1.0).”
Thank you very much.
Best regards,
Dr. Tzung-Chien Hsieh